# IL-20 Cytokines Are Involved in the Repair of Airway Epithelial Barrier: Implication in Exposure to Cigarette Smoke and in COPD Pathology

**DOI:** 10.3390/cells12202464

**Published:** 2023-10-16

**Authors:** Olivia Barada, Sophie Salomé-Desnoulez, Fahima Madouri, Gaëtan Deslée, Christelle Coraux, Philippe Gosset, Muriel Pichavant

**Affiliations:** 1Institut Pasteur de Lille, Centre d’Infection et d’Immunité de Lille; Université Lille Nord de France; Centre National de la Recherche Scientifique UMR 9017; Institut National de la Santé et de la Recherche Médicale U1019, 59019 Lille, France; olivia.barada@gmail.com (O.B.); fahima.madouri@cnrs-orleans.fr (F.M.); philippe.gosset@pasteur-lille.fr (P.G.); 2Institut Pasteur de Lille, Université de Lille, CNRS UMR9017, Inserm U1019, CHU Lille, US 41—UAR 2014—PLBS, 59000 Lille, France; sophie.salome-desnoulez@ibl.cnrs.fr; 3Service de Pneumologie, Centre Hospitalier Universitaire de Reims, 51092 Reims, France; gdeslee@chu-reims.fr; 4Institut National de la Santé et de la Recherche Médicale, UMR-S 1250, Université de Reims Champagne-Ardenne (URCA), SFR Cap-Santé, 51100 Reims, France; christelle.coraux@univ-reims.fr

**Keywords:** airway epithelial cells, IL-20 cytokine family, COPD and epithelial repair

## Abstract

Background: Dysregulated inflammation as seen in chronic obstructive pulmonary disease (COPD) is associated with impaired wound healing. IL-20 cytokines are known to be involved in wound healing processes. The purpose of this study was to use ex vivo and in vitro approaches mimicking COPD to evaluate the potential modulatory role of interleukin-20 (IL-20) on the inflammatory and healing responses to epithelial wounding. Methods: The expression of IL-20 cytokines and their receptors was investigated in lung-derived samples collected from non-COPD and COPD patients, from mice chronically exposed to cigarette smoke and from airway epithelial cells from humans and mice exposed in vitro to cigarette smoke. To investigate the role of IL-20 cytokines in wound healing, experiments were performed using a blocking anti-IL-20Rb antibody. Results: Of interest, IL-20 cytokines and their receptors were expressed in bronchial mucosa, especially on airway epithelial cells. Their expression correlated with the disease severity. Blocking these cytokines in a COPD context improved the repair processes after a lesion induced by scratching the epithelial layer. Conclusions: Collectively, this study highlights the implication of IL-20 cytokines in the repair of the airway epithelium and in the pathology of COPD. IL-20 subfamily cytokines might provide therapeutic benefit for patients with COPD to improve epithelial healing.

## 1. Introduction

Chronic obstructive pulmonary disease (COPD) is a heterogeneous disease characterized by progressive airflow obstruction and airway inflammation, variously associating airway epithelium remodeling, lung parenchymal destruction, systemic inflammation, comorbidities and higher susceptibility to pulmonary infections [1]. This disease is mostly related to chronic exposure to aeropollutants, including tobacco smoke and indoor/outdoor air pollution. COPD represents a major worldwide health topic due to its morbidity, mortality and its large consumption of health care resources, with current treatments being of limited effect [2]. The development of new therapeutic strategies requires a better understanding of the mechanisms underlying the physiopathology of COPD. 

Airway epithelium is essential for the orchestration of the local inflammatory and immune responses and plays a central barrier function associated with the mucociliary apparatus controlling the elimination of inhaled substances within the lung tissue [3]. Many aeropollutants, including tobacco smoke, airborne particulates, noxious gases, infectious agents and allergens can alter the airway epithelium [4]. Remodeling of large and small airways, including squamous metaplasia, ciliated cell defect [5], subepithelial fibrosis, goblet cell hyperplasia, mucus gland hypertrophy and hyperplasia of bronchial smooth muscle, have been shown in COPD. Desquamation and cell death also occur within airway epithelium and require a complex repair mechanism to restore its integrity. This repair process includes spreading and migration of cells neighboring the wound, proliferation and progressive re-differentiation until a complete regeneration of a pseudostratified mucociliary epithelium [6]. However, the mechanisms involved in this repair process at homeostasis as well as the physiopathologic mechanisms responsible for airway remodeling during COPD are not fully understood.

Among factors involved in epithelium repair, IL-22, a cytokine belonging to the IL-20 subfamily, has been shown to stimulate cell proliferation, the production of antimicrobial proteins and pro-inflammatory mediators [7]. Altogether, these functions strengthen epithelial barrier function. IL-22 was reported to control the communication between leukocytes and epithelial cells, and thereby enhance innate defense mechanisms and tissue repair processes at epithelial surfaces including the airways [8]. In addition to IL-22, the IL-20 subfamily comprises IL-19, IL-20, IL-24 and IL-26, and is part of the larger IL-10 family of cytokines [9]. These cytokines are mainly produced by hematopoietic cells, but they signal through receptors primarily expressed in epithelial tissues including the airways [10]. IL-20 subfamily cytokines signal through heterodimeric receptors composed, for IL-22, of the IL-10 receptor β-subunit (IL-10RB) associated with IL-22RA1, and for IL-19, IL-20 and IL-24 (IL-20 cytokines) of IL-20RB paired with either the IL-20 receptor α-subunit (IL-20RA) or IL-22RA1 [11]. A soluble receptor for IL-22 has been identified as IL-22-binding protein (IL-22BP), a protein with important homology to IL-22RA1 [12]. 

IL-20 cytokines have a broad range of functions and are involved in a variety of immune and non-immune processes in the body [13]. In the skin, IL-20 cytokines are induced during wound healing, and contribute to this process by inducing inflammation, re-epithelialization and remodeling [14]. Moreover, in psoriatic skin, IL-20Rb is highly upregulated [15]. Dysfunctional regulation of IL-20 cytokines could lead to uncontrollable wound healing in psoriasis which could be a contributing factor to the pathogenesis of this disease, although their role in airway mucosa was unknown. 

We recently observed that IL-20 cytokines can be induced in the lung by environmental factors, including cigarette smoke and pathogens [16]. Based on these data, we hypothesized that IL-20 cytokines could be involved in the physiopathology of COPD and the abnormal wound repair of the airway epithelium observed in this pathology. To assess this assumption, we first studied the expression of IL-20 in lung sections and epithelial cells from non-COPD and COPD patients, as well as human cells exposed to cigarette smoke, and then the role of IL-20 cytokines in the inflammatory response and during wound healing of airway epithelium.

## 2. Materials and Methods

### 2.1. Human Samples 

Human lung samples from female and male patients scheduled for lung resection for cancer (Reims University Hospital, Reims, France) were prospectively recruited (*n* = 46) according to the standards established and approved by the Institutional Review Board of the University Hospital of Reims, France (IRB Reims-CHU-20110612, approval date: 12 June 2011) (Table 1). Upon inclusion, age, sex, smoking status and pulmonary function test results were recorded. Ex-smokers were defined as having had a withdrawal longer than 12 months. Samples were collected distant from the tumor, were fixed in formalin and embedded in paraffin for immunohistochemical analysis. The paraffin-embedded tissues were obtained from samples of the Tissue Bank of the Reims University Hospital Biological Resource Collection No. DC-2008-374 declared at the Ministry of Health, according to the French law for the utilization of tissue samples for research. Surgically resected tissues were collected after obtaining informed consent from patients (document provided). Sex has been taken into account in performing analysis.

### 2.2. Experimental Animal Model

Six- to eight-week-old male wild-type (WT) C57BL/6 (H-2Db) mice were purchased from Janvier (Le Genest-St.-Isle, France). All animal work conformed to the guidelines of the Animal Care and Use Committee of Nord Pas-de-Calais (agreement no. AF 16/20,090). Mice were maintained in a temperature-controlled (23 °C) facility with a strict 12 h light/dark cycle and with food and water provided ad libitum. Mice were daily exposed to cigarette smoke (CS) for 12 weeks (5 days/week for 12 weeks) to induce COPD-like pathogenesis [17]. Research cigarettes 3R4F were obtained from the University of Kentucky Tobacco and Health Research Institute (Lexington, KY, USA). The control group was exposed to ambient air (air mice). Mice were sacrificed at the end of the exposure by cervical dislocation and the lungs and tracheas of the mice were collected. 

### 2.3. Lung Immunohistochemistry

For histopathology, lungs of mice were fixed by inflation and immersion in paraformaldehyde (PFA; 4%) and embedded in paraffin. To evaluate airway inflammation, lung sections (4-µm thick) were stained with hematoxylin and eosin. On these slides, we measured both lung remodeling and inflammation through the use of lung injury scoring.

For immunohistochemistry, paraffin-embedded lung sections were deparaffinized into xylene (Acros Organics, Thermo Fisher, Les Ulis, France) and rehydrated with successive baths of alcohol and water. The unmasking of epitopes was carried out in pH 6.0 phosphate buffer (Na_2_HPO_4_, 12H_2_O) for 15 min at 90 °C. Primary antibodies against IL-19 (ab154187 (Abcam, Canada)), IL-20 (orb13501 (Biorbyt, Cambridge, United Kingdom)), IL-24 antibody (orb228807 (Biorbyt, Cambridge, United Kingdom)) and IL-20Rb (clone 20RNTC (Invitrogen, Thermo Fisher, France)) as well as isotype controls were all used at 0.01 mg/mL. The Expose Mouse and Rabbit Specific HRP/DAB Detection IHC Kit (Abcam, Canada) was used according to the manufacturer’s recommendation for primary antibodies detection. DAB was used for revelation. Counterstaining was performed on lung sections with Hematoxylin (Interchim, Montluçon, France). 

### 2.4. BEAS-2B Cell Cultures

BEAS-2B cells were obtained from the ATCC (Manassas, VA, USA). BEAS-2B cells were maintained in airway epithelial cell growth medium (Promocell, Heidelberg, Germany). Cell lines were cultivated on culture plates coated with human type I collagen G matrix (Biochrom, Berlin, Germany). When cells were confluent, BEAS-2B cells were recovered at 4 h post-treatment in order to collect mRNA.

### 2.5. Mouse Tracheal Epithelial Cells (mTEC) Cell Cultures

The mouse tracheas were digested in DMEM F12 medium (Gibco, Fisher Scientific, Illkirch, France) + 1% Penicillin-Streptomycin (Pen/Strep; Gibco, Fisher Scientific, Illkirch, France) + pronase (1.6 mg/mL, Roche, Fisher Scientific, Illkirch, France) for 1 h at 37 °C. The reaction was stopped with Fetal Calf Serum (Gibco, Fisher Scientific, Illkirch, France). The epithelial cells were recovered by vortexing and cultured in DMEM F12 medium supplemented with 2% Ultroser G (Gibco, Fisher Scientific, Illkirch, France), antibiotics and 2 mM L-Glutamine (PALL Life Sciences, Cytiva, Marlborough, MA, USA). Cells were cultivated on culture plates coated with human collagen G matrix (Biochrom, Berlin, Germany). When cells were confluent, mTEC were recovered at 4 h post-treatment in order to collect mRNA.

### 2.6. Cytokine Measurement by Individual ELISA System Kit

Human hIL-19, hIL-20 and hIL-24 (R&D Systems, DY1035, DY1102 and DY1965, respectively), were determined in BEAS-2B cell lysates (dilution ½) by enzyme-linked immunosorbent assay (ELISA), using commercial kits according to the manufacturer’s recommendations (R&D Systems, Noyal Châtillon sur Seiche, France).

### 2.7. RNA Isolation and Quantitative RT-PCR

Total RNA was isolated from cells and the lungs of mice using TRIzolR Reagent (Ambion, Austin, TX, USA) and quantified by NanoVue Plus (Healthcare Bio-sciances AB, Uppsala, Sweden). Reverse transcription was performed with a High Capacity cDNA Reverse Transcription Kit (Applied Biosystems, Waltham, MA, USA) according to manufacturer’s instructions. cDNA was subjected to quantitative PCR (QuantStudio 12K Flex Applied Biosystems, Thermo Scientific, Waltham, MA, USA) using specific primers for mouse (mTEC cells) or for human (BEAS-2B cells) (Table 2 and Table 3, respectively) (Eurofins Genomics, Ebersberg, Germany). Relative transcript expression of a gene is given as 2^−ΔΔCt^ by using mouse gapdh and human actin as a housekeeping gene, respectively.

### 2.8. Cigarette Smoke Extract

Cigarette smoke extract (CSE) was prepared according to the method described by Blue and Janoff [18]. The smoking apparatus consisted of a 50-mL syringe to which a cigarette was attached. CSE was prepared by drawing 50 mL of cigarette smoke through the filter into the syringe and then slowly bubbling the smoke into 10 mL of basal Airway Epithelial Cell Medium (PromoCell, Heidelberg, Germany). Two Kentucky research cigarettes 3RF4 were smoked per 10 mL of medium. The final solution was filtered through 0.2 μm filters and used immediately at 1/50 dilution.

### 2.9. Wound Healing Closure Assay

Confluent BEAS-2B cells were prepared in 24-well plates and pretreated or not with Mitomycin C (25 µg/mL) for 1 h before the injury. Then, the injury line was made with a pipette tip and detached cells were discarded by rinsing with PBS. Medium was changed for basal epithelial cell growth medium (without growth factors, Promocell, Heidelberg, Germany). Recombinant cytokines IL-19, IL-20, IL-22 and IL-24 were added to the culture medium at 10 ng/mL concentration. In some experiments, the neutralizing anti-IL-20RB antibody (clone 20RNTC) or a control isotype was added to the culture at 5 µg/mL before stimulation. Defined zones of the injury line were photographed (×120 magnification) at the indicated times (0, 24, 48 and 72 h). The cell-free surface in the wells was quantified by using Image J software (https://imagej.net/ij/index.html) according to the following method: the free area at t = 0 was delimited and was reported at 100% for each condition. Subsequently, for each point, the remaining free area was again delimited and expressed as a percentage of the remaining area. The wound healing closure assay was performed using a Zeiss ObserverZ.1 microscope system (Zeiss, Jena, Germany) with a 10× dry lens (Plan Neofluar 10×/0.3 Ph1) and equipped with a thermostat-controlled chamber regulated to 37 °C and 5% CO_2_. A DIC picture was taken every 30 min for 72 h by a sCMOS camera Prime 95B (Photometrics) and processed with ZEN blue 2.3 software (Zeiss, Jena, Germany). Quantitative analysis of the wound closure rates was performed with the software Fiji (https://fiji.sc/; NIH, Bethesda, MD, USA) delimiting the borders of the wound and determining the area of the free surface.

### 2.10. Transcriptomic Analysis

The murine right lung lobe was collected and stored at −80 °C until RNA extraction. The RNA was extracted using the RNeasy Mini Kit (QIAGEN, Hilden, Germany) according to the manufacturer’s instructions. The isolated RNA was then quantified using a Nanodrop 1000 Spectrophotometer (Thermo Scientific, Waltham, MA, USA) and the quality checked using the Agilent 2100 Bioanalyser (Agilent Technologies, Santa Clara, CA, USA).

Preparation of RNA for RNAseq: Total RNA quality and concentration were assessed by using a Bioanalyzer RNA nano 6000 kit (Agilent) and the RiboGreen assay (Thermofisher Scientific, Waltham, MA, USA), respectively. Strand-specific RNA-Seq libraries were prepared with the Truseq stranded mRNA sample prep kit (Illumina, San Diego, CA, USA) following the manufacturer protocol, including mRNA isolation thanks to oligo dT coated beads. After controlling the libraries’ quality on 2100 Bioanalyzer Instrument (Agilent, ref G2938B), an equimolar pool was prepared and paired-end sequenced (2 × 150 bp) on 4 lanes of the HiSeq 4000 platform (Illumina). The sequencing run was analyzed with the Illumina CASAVA pipeline with demultiplexing based on sample-specific barcodes. 

Reads cleaning: Raw reads were cleaned using cutadapt software v1.15 [19] to remove adapter in 3′ of the sequences. The reads were then processed with prinseq software v0.20.4 [20] in order to trim the first 15 bases of each read, low quality bases (phred score < 30), undetermined nucleotides and portion of sequences whose mean phred score over 7 bases was less than 30 in 3′ of the reads. Reads shorter than 60 bases and reads remaining unpaired after the trimming step were discarded.

Differential expression analysis: Cleaned reads were mapped on the CDS of the reference genome of *Mus musculus* (GCF_000001635.26 GRCm38.p6) using the script rsem-calculate-expression provided by the RSEM pipeline v1.3.0 [21] with bowtie2 aligner and default options. The count matrices were then merged by comparison using rsem-generate-data-matrix, and for each comparison the resulting merged matrix was processed by rsem-run-ebseq with default options in order to perform a differential expression analysis using the EBSeq R package [22]. The differential expression analyses were finally filtered on the FDR value with rsem-control-fdr using 4 different cutoff values: 0.05, 0.01, 0.005 and 0.001. 

### 2.11. Statistical Analysis

All the experiments were repeated at least 3 times. Results are expressed as the mean ± SEM. Mann–Whitney U analysis and *t*-test were performed to determine significant differences between groups using GraphPad Prism software version 5.00. Statistically significant differences were defined as: * *p* < 0.05, ** *p* < 0.01 and *** *p* < 0.001.

## 3. Results

### 3.1. The IL-20 Cytokines Are Increased in the Airway Mucosa of COPD Patients

We investigated the expression pattern of IL-20 cytokines and related receptor subunits on lung sections from COPD patients compared to current non-COPD smokers, non-smokers and ex-smokers (Table 1). We first evaluated the production of IL-19, IL-20 and IL-24 cytokines (Figure 1A and Table 4). Our results show the production of IL-20 cytokines, mainly of IL-19, which is localized in human airway epithelium from non-smokers. Exposure to CS seems to increase the expression of IL-19, IL-20 and IL-24 in airway epithelium, as seen in smokers. Interestingly, smoking cessation does not alter IL-20 cytokine production as compared with active smokers. 

We also analyzed the production of IL-20 cytokines on lung sections from COPD patients with spirometric GOLD I and II stages. In addition to the significant airway and mesenchymal remodeling associated with the presence of inflammatory infiltrates, the production of IL-20 cytokines was increased in the airway epithelium (mostly of IL-20 and IL-24) and in the subepithelial tissue, particularly within infiltrating immune cells. Moreover, the subepithelial staining was more pronounced in COPD patients than in smokers, whereas this is not the case in airway epithelium. We did not see any difference between GOLD I and II COPD patients. Some endothelial and immune cells were also positive for IL-20 cytokine staining.

Overall, our observations suggest that IL-20 cytokines are expressed both in the epithelium and infiltrating immune cells, and that their expression might be modulated by CS exposure and/or the COPD-associated disease. 

We next analyzed the expression of the different subunits of IL-20 receptors, namely IL-20Ra, IL-20Rb and IL-22Ra (Figure 1B and Table 4). Our results show that the IL-20Ra subunit is highly expressed in airway mucosa, particularly in airway epithelial cells from non-COPD patients, and does not appear to be modulated by CS exposure and/or the disease.

In contrast, the IL-22Ra subunit is very weakly expressed in all donors, whether smoking or not. However, the production of IL-22Ra was detected in association with lung inflammatory infiltrate in some subjects, suggesting a potential role of this subunit in the context of inflammation. Regarding IL-20Rb, this subunit is expressed at a low level in non-smoking patients in the different cell types including BEC within the airway epithelium. CS seems to induce an increase in the expression of IL-20Rb. Smoking cessation is associated with a decrease in IL-20Rb expression, particularly within the subepithelial area. COPD patients also exhibited an increased expression of the IL-20Rb subunit. The IL-20Ra subunit remained highly expressed and did not show any modulation in GOLD I and II COPD patients. Finally, the IL-22Ra subunit remains very weakly expressed in this pathological context, whereas our results suggest the involvement of the IL-20Rb subunit in the pathology. In addition, we also observed some staining for the three subunits on endothelial cells and some immune cells.

### 3.2. The IL-20 Cytokine Members Are Expressed in the Lungs of Mice Chronically Exposed to Cigarette Smoke

We next evaluated the expression pattern of IL-20 cytokines and their receptors in the lungs of mice chronically exposed to CS, mimicking COPD pathology, in comparison to mice exposed to ambient air. 

In control conditions (air mice), the expression of IL-19 and IL-24 was evaluated by immunohistochemistry and was mainly detected in the airway epithelium, whereas IL-20 was nearly undetectable (Figure 2A, left). When mice were chronically exposed to CS, an increased expression of IL-19, IL-20 and IL-24 was seen, especially in airway epithelial cells but also in pneumocytes and some alveolar macrophages. These data confirmed that exposure to CS strongly induced the production of IL-20 cytokines in mouse lungs. We also analyzed the expression of the three subunits of IL-20 cytokine receptors (Figure 2A, right). The data showed, that, as in human lungs, IL-20Ra is the most abundantly expressed subunit in the lungs of air mice, particularly in the airway epithelium. IL-22Ra is also expressed in epithelial cells, but more weakly than IL-20Ra whereas IL-20Rb was undetectable. When mice are exposed to CS, the expression of both IL-20Rb and IL-22Ra is higher within airway epithelial cells.

We next investigated the level of mRNA transcripts by qPCR in total lung tissue (Figure 2B). A significant increase in the mRNA transcripts encoding for *il-19* and *il-24* was observed in mice exposed to CS. A trend for an increase in *il-20* transcript was also observed. Concerning the expression of *il-20rb* mRNA in the total lung, a significant increase in expression is observed in the lungs of smoking mice.

These data are in line with the previous results observed in human patients, thus showing a modulation of the expression of IL-20 cytokines and of the IL-20Rb subunit potentially related to chronic exposure to CS.

To elucidate the potential role of these IL-20 cytokines in the pathology of COPD, we next performed some transcriptomic experiments on total lung extracts. As shown in Figure 2C, exposure to CS was associated with modification in the expression of genes involved in airway remodeling and wound healing.

### 3.3. In Vitro Exposure to Cigarette Smoke Triggers the Expression of IL-20 Cytokines in Human and Murine Airway Epithelial Cells

Since IL-20 cytokines and the IL-20Rb subunit were mainly expressed by airway epithelial cells, we then translated our observations onto BEAS-2B cells, a human airway epithelial cell line. In Figure 3A, exposure to CSE significantly increased *il-24* and *il-20rb* mRNA levels, whereas transcripts for *il-19* and *il-20* tended to be upregulated. In contrast, we did not observe any variation of IL-19 and IL-20 levels in BEAS-2B cell supernatants, except for IL-24 when cells were exposed to CSE (Figure 3B).

We also investigated the *il-20* cytokines and *il-20rb* subunit expression in murine airway epithelial cells (Figure 4A). Exposure to CSE significantly increased *il-19* mRNA levels, whereas the increase in transcripts for *il-20* and *il-24* was not significant. Transcripts for *il-20rb* were not impacted by CS exposure.

We next performed some transcriptomic experiments on murine airway epithelial cells exposed or not to CSE. As shown in Figure 4B, exposure to CS was associated with modification in the expression of genes involved in airway remodeling and wound healing.

### 3.4. IL-20 Cytokines Play a Role in Epithelial Repair

In the literature, it has been demonstrated that IL-20 cytokines play a role in skin epithelial repair at homeostasis and in the physiopathology of skin diseases [13,23,24]. Since their roles remain unknown in the lungs, we looked at whether IL-20 cytokines had an impact on wound healing by focusing first on BEAS-2B cell proliferation and migration and the implication of IL-20Rb in this context (Figure 5). 

In control conditions, we observed that IL-19, IL-20 and IL-24 cytokines significantly accelerate cell repair (Figure 5A). We performed some kinetic approaches to evaluate how exogenous IL-19, IL-20 and IL-24 cytokines could affect airway epithelial wound healing (Figure 5B). The kinetic study revealed that addition of recombinant IL-19, IL-20 and IL-24 accelerated epithelial repair. The treatment with IL-19, IL-20 or IL-24 increased the cell repair. Interestingly, the use of anti-IL20Rb antibody in this context seems to decrease cell repair, suggesting the importance of IL-20 cytokines in this process (Figure 5B,C). Finally, we tested whether IL-20 cytokines had a role in cell migration or proliferation. To this end, we treated BEAS-2B cells with mitomycin C, an antimitotic drug [25] (Figure 5D). Thus, we found that when cells were treated with mitomycin C and IL-20 cytokines we did not observe any differences in repair. This leads us to the conclusion that IL-20 cytokines are involved in the process of cell proliferation but not in cell migration.

Next, we wanted to analyze the impact of CSE on wound healing of BEAS-2B cells (Figure 6). In contrast to control conditions, when cells are treated with CSE the anti-IL20Rb antibody alone seems to accelerate epithelial cell repair (Figure 6A). In addition to this observation, when we performed the kinetic approach (Figure 6B), we observed that IL-19, IL-20 or IL-24 cytokines seem to slow down the healing process. However, when the anti-IL-20Rb antibody is added to these conditions, we observe a repair that seems to be accelerated (Figure 6B,C). These results suggest that in the presence of CSE, IL-20 cytokine signaling is altered compared to untreated conditions. Taken together, these data suggest that the anti-IL20Rb blocking antibody favors BEAS-2B cell repair only in the context of an exogenous stress, such as CSE. Finally, when cells are treated with mitomycin C, the same results are observed as in control conditions, which seems to confirm the role of IL-20 cytokines in cell proliferation (Figure 6D).

In summary, these results show that IL-20 cytokines can promote the repair of airway epithelium in control conditions. Moreover, the results obtained tend to show that these cytokines have a role in cell proliferation, as in the presence of mitomycin C cell proliferation is inhibited. In contrast, after exposure to CSE, these cytokines can have a negative role on epithelial repair by slowing it down. Interestingly, we demonstrated here that neutralization of the IL-20Rb subunit was able to partially restore epithelial repair when cells were exposed to cigarette smoke, providing evidence that cell signaling induced by IL-20 family cytokines is involved in this process at the respiratory level. Furthermore, the anti-IL-20Rb blocking antibody could also represent a potential therapeutic tool to promote epithelial repair in the context of chronic respiratory disease such as COPD.

## 4. Discussion

COPD is a heterogeneous disorder characterized by irreversible airway obstruction and lung remodeling [1]. The immune mechanisms involved in the development of COPD are not yet fully understood. In the present study, we demonstrated that IL-20 cytokines are highly expressed in the airway epithelium after exposure to CS, and this effect is amplified in COPD patients as compared with smokers. This led us to look at what was happening in the context of this chronic pathology mainly caused by cigarette smoke exposure, namely during the repair of the lung epithelial barrier.

We first investigated the expression of IL-20 cytokines and their receptors on COPD lung sections by immunohistochemistry analysis, and we showed that the expression of IL-20 in the lungs is higher in non-COPD smokers compared to non-smoking controls. Nevertheless, the development of COPD is associated with an increased expression of these cytokines compared to non-COPD smokers. Since GOLD II patients exhibited higher expression of IL-20 cytokines, and especially IL-19, than GOLD I patients, we can suspect that the levels of these cytokines are associated with the severity of the disease, although it would require the inclusion of a sufficient number of patients, including patients with GOLD III and IV, to test this hypothesis. Interestingly, the expression of IL-20 cytokines seems to be associated to an increased level of IL-20RB, whereas the IL-20RA subunit is highly expressed in all groups and IL-22Ra remains at a low level. This suggests that IL-20RB might be an important marker of the function of IL-20 cytokines in this context. Aberrant IL-20 expression has been implicated in numerous inflammatory diseases, including psoriasis [13]. Indeed, previous studies have demonstrated that IL-20 cytokines and their receptors were overexpressed in keratinocytes of the lesional skin of psoriasis and spongiotic dermatitis [26]. However, the role of IL-20 in these pathologies is not fully understood. When examining disease-driven changes of IL-20 and its cognate receptor subunits in skin from healthy human subjects, atopic dermatitis (AD) patients and murine AD models, the authors highlighted a role of IL-20 signaling in the pathophysiology of AD, thus forming a new basis for the development of a novel antipruritic strategy via interrupting IL-20 epidermal pathways [27]. These data suggest that IL-20 cytokines and their receptors might be therapeutic targets against mucosal chronic inflammation [28], including both the skin and the lung. Since the expression of these cytokines seems to be related to the gravity of the pathology, IL-20 cytokines might be considered as new biomarkers of severity, although this would need to be confirmed in COPD patients.

To continue the parallel between the skin and the lungs, it has been shown that overexpression of these cytokines was correlated with keratinocyte proliferation through an autocrine loop [26]. Because IL-20 cytokines are involved in the promotion of proliferation of epithelial cells, they are also linked to barrier repair, and their receptors are very often expressed on epithelial cells. Therefore, IL-20 cytokines could act through an autocrine loop. Since lung epithelial cells also express the receptors involved in IL-20 signaling, our results suggest that aberrant expression of IL-20 cytokines might be associated with an alteration of the lung epithelial barrier. Physiological airway epithelial wound repair is a complex, highly orchestrated process, presenting numerous points where dysregulation may occur, leading to the development of several pulmonary disorders, including COPD. In their study, Perotin et al. [5] demonstrated that the mean speed of epithelial wound closure decreased with the severity of COPD, using primary airway epithelial cells obtained from patients that were clinically, functionally and histologically characterized. They also showed that the speed of wound closure of bronchial and bronchiolar epithelial cells obtained from the same patients was strongly associated. Repair of airway epithelium after injury is critical for the maintenance of the airway epithelial structure and function. In our study, we showed that IL-20 cytokines accelerate wound closure. Interestingly, the effect of recombinant cytokines is abrogated in presence of CSE, and blocking the autocrine function of IL-20 cytokines induced by CS exposure using the anti-IL20Rb antibody restores an appropriate wound closure. These results suggest that IL-20 cytokines that are produced in higher quantity in the context of CS exposure and/or by COPD patient airways could contribute to the defective repair capacity of the airway epithelium. Moreover, we recently demonstrated that IL-20 cytokines alter the epithelial barrier in the context of virus-induced exacerbation of COPD in mice by favoring the impact of the infection on the disorganization of intercellular junction protein [29]. This suggests that blocking IL-20 signaling could represent a potential therapeutic approach for COPD patients. Anti-IL-20 monoclonal antibodies have been evaluated as clinical candidates for the treatment and/or prevention of psoriasis, rheumatoid arthritis, atherosclerosis, osteoporosis, stroke and also bacterial lung infections [30]. The anti-IL-20 antibody neutralizes IL-20 signaling, an effect associated with a decreased production of TNF-α, IL-1 and IL-6 signaling in vivo. Blocking anti-IL20Rb monoclonal antibodies could be also of high interest in COPD pathogenesis, since it can neutralize the effect of IL-19, IL-20 and IL-24, these cytokines being induced by CS exposure.

COPD pathogenesis is also characterized by a strong airway remodeling. MMP-2 and MMP-9 have been shown to be involved in this process [31]. These proteases could play a role in the epithelial repair process [32]. During epithelial repair, MMP-9 is secreted by transdifferentiated cells and degrades focal adhesion at the rear of the cell [33]. Moreover, inactivation of MMP-9 has been shown to decrease the migration of airway epithelial cells [34]. Previous studies showed an association between MMP-2 level and airway obstruction. Perotin et al. found that MMP-2 level decreased in severe COPD patients, which may play a role in impaired airway epithelium repair [5]. Other studies also demonstrated elevated blood concentrations of MMP-9 in COPD patients, suggesting that MMP-9 could even represent a novel biomarker that identifies a subset of individuals with COPD with an inflammatory phenotype who are at increased risk for acute exacerbation [35]. Interestingly, upregulation of MMP-2 and MMP-9 have been shown after IL-20 stimulation [36]. In a previous study, we have demonstrated that expression of IL-20 cytokines was induced within epithelial cells by bacterial and viral respiratory infections that often cause COPD exacerbation. Thus, the crosstalk between MMP-9 and IL-20 cytokines in COPD is ambiguous and needs to be further explored.

## 5. Conclusions

Altogether, our data showed that exposure to CS induces the expression of IL-20 cytokines and promotes the expression of the IL-20Rb subunit, a process exacerbated during the development of COPD. In this pathological context, this modulation is associated with an alteration of the airway epithelium wound healing, whereas these IL-20 cytokines do not seem to modulate the production of inflammatory markers. Our data also suggest that blocking the IL-20 cytokine pathways by using anti-IL-20Rb antibody might be of interest for the treatment of COPD patients.

## Figures and Tables

**Figure 1 cells-12-02464-f001:**
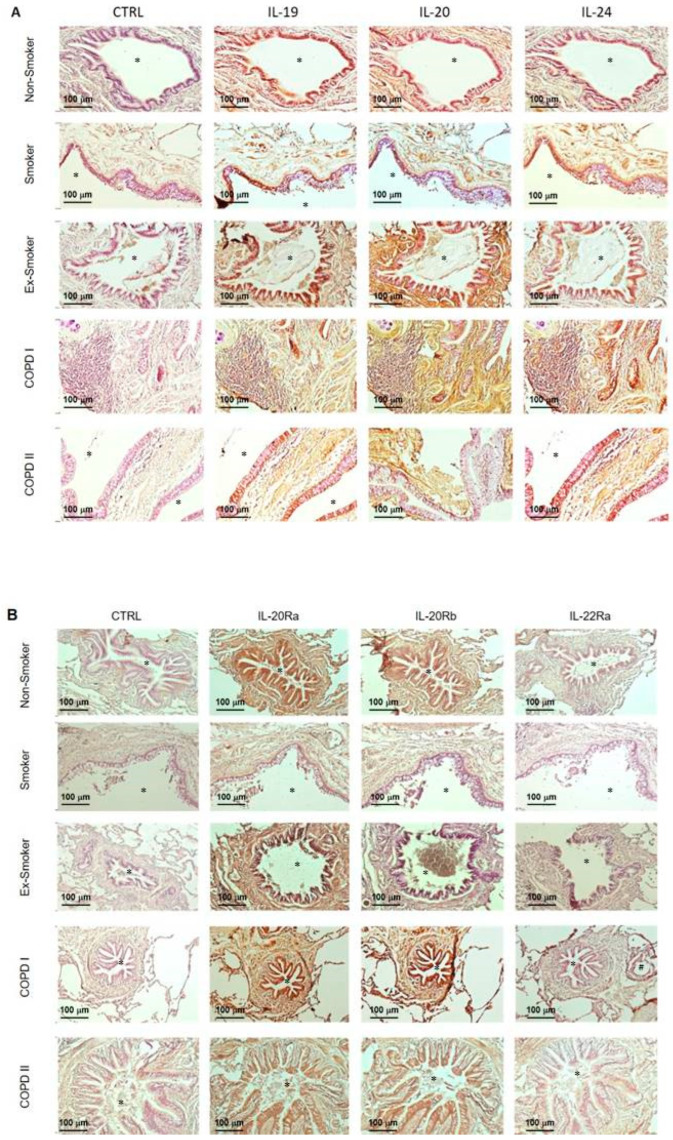
Expression of IL-20 cytokine family members and their receptors in the airways of patients with and without COPD. Representative pictures of immunohistological staining for each group of patients (magnification x100). (**A**) Staining of cytokines IL-19, IL-20 and IL-24 in controls, non-smokers, ex-smokers, current smokers and GOLD I and II COPD patients. (**B**) Staining of IL-20Ra, IL-20Rb and IL-22Ra receptors in the same conditions as mentioned in part A. * bronchial lumen, ^#^ vascular lumen.

**Figure 2 cells-12-02464-f002:**
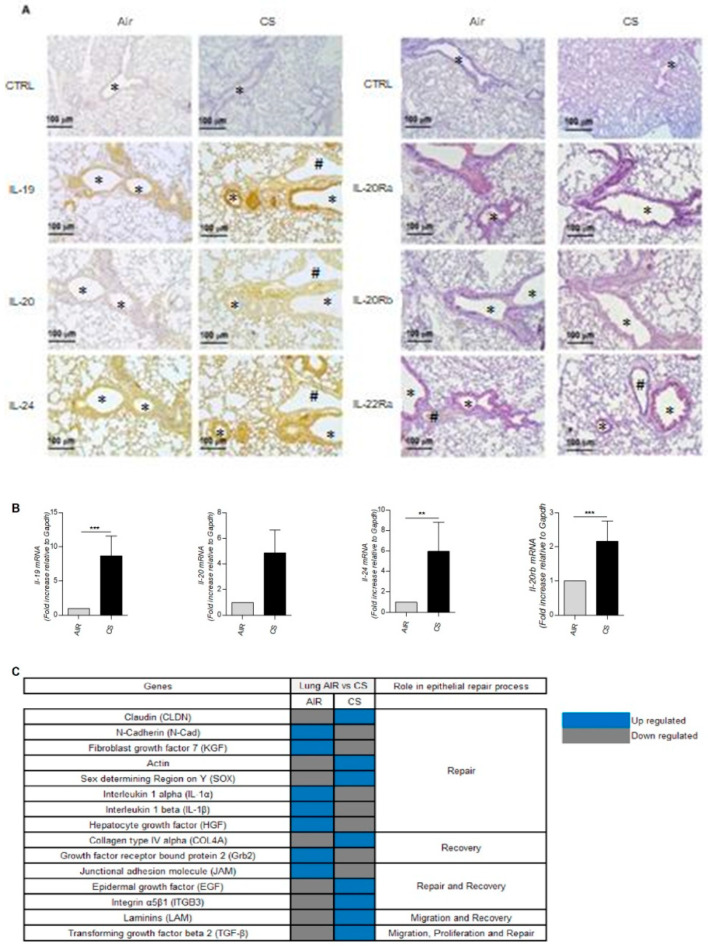
Expression of IL-20 cytokine family members and their receptors in the lungs of mice chronically exposed to air or CS. Lungs from mice exposed to air and chronically to CS were collected. (**A**) Expression of cytokines including IL-19, IL-20 and IL-24, and corresponding receptor subunits IL-20Ra, IL-20Rb and IL-22Ra was evaluated on lung sections by immunohistochemistry (magnification ×100). *: bronchial lumen, ^#^: vascular lumen. (**B**) mRNA levels encoding for *il-19*, *il-20, il-24* and *il-20rb* were evaluated by RT-qPCR in total lung tissue extracts. Results are expressed as 2^−ΔΔCt^ by using mouse *gapdh* as a housekeeping gene. (**C**) Expression of genes by mRNA-Seq involved in cell repair in total lung extracts. Results are expressed as mean ± SEM and statistical analyses were performed by Mann–Whitney U analysis and *t*-test in comparison with lungs not exposed to CS. ** *p* < 0.01 and *** *p* < 0.001. Independent experiments have been performed between 3–7 times.

**Figure 3 cells-12-02464-f003:**
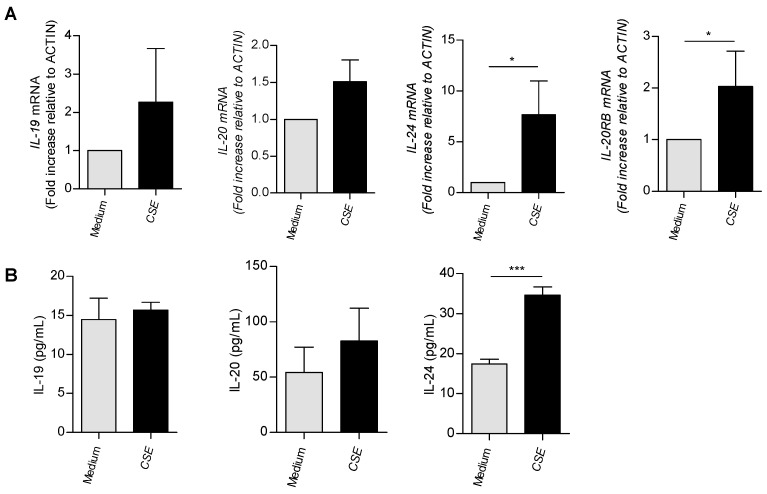
mRNA expression and protein secretion of IL-20 cytokines and their receptors in BEAS-2B cells exposed or not to CSE. BEAS-2B cells were cultured and exposed or not to CSE. Cells for mRNA extraction and supernatants for ELISA assays were collected at 4h and 24 h post-CSE exposure, respectively. (**A**) *il-19*, *il-20, il-24* and *il-20rb* mRNA levels were evaluated by RT-qPCR. Results are expressed as fold increase compared to cells not exposed to CSE and using expression of *actin* as a housekeeping gene. (**B**) Assessment of secretion of IL-19, IL-20 and IL-24 by ELISA (pg/mL). Results are expressed as mean ± SEM and statistical analyses were performed by Mann–Whitney U analysis and *t*-test in comparison to cells not exposed to CSE. * *p* < 0.05, and *** *p* < 0.001. Independent experiments have been performed between 4–8 times.

**Figure 4 cells-12-02464-f004:**
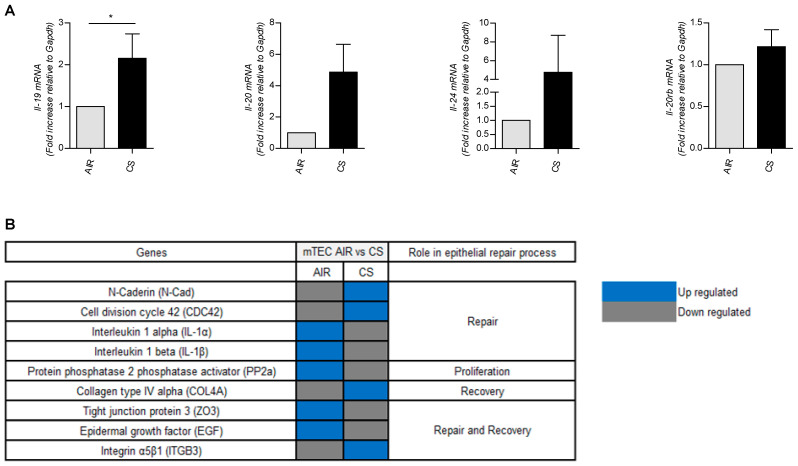
Modulation by CS extract of mRNA expression for IL-20 cytokines, their receptors and genes involved in cell repair in mTEC cells. mTEC cells were cultured after exposure of mice to air or CS. (**A**) *il-19, il-20*, *il-24* and *il-20rb* mRNA levels were evaluated by RT-qPCR. Results are expressed as fold increase compared to cells not exposed to CS and using expression of *gapdh* as a housekeeping gene. (**B**) Expression of genes by mRNA-Seq involved in cell repair by mRNA-Seq in mTEC cells from mice exposed or not to CS. Results are expressed as mean ± SEM and statistical analyses were performed by Mann–Whitney U analysis and *t*-test in comparison with cells not exposed to CSE. * *p* < 0.05. Independent experiments have been performed between 4–8 times.

**Figure 5 cells-12-02464-f005:**
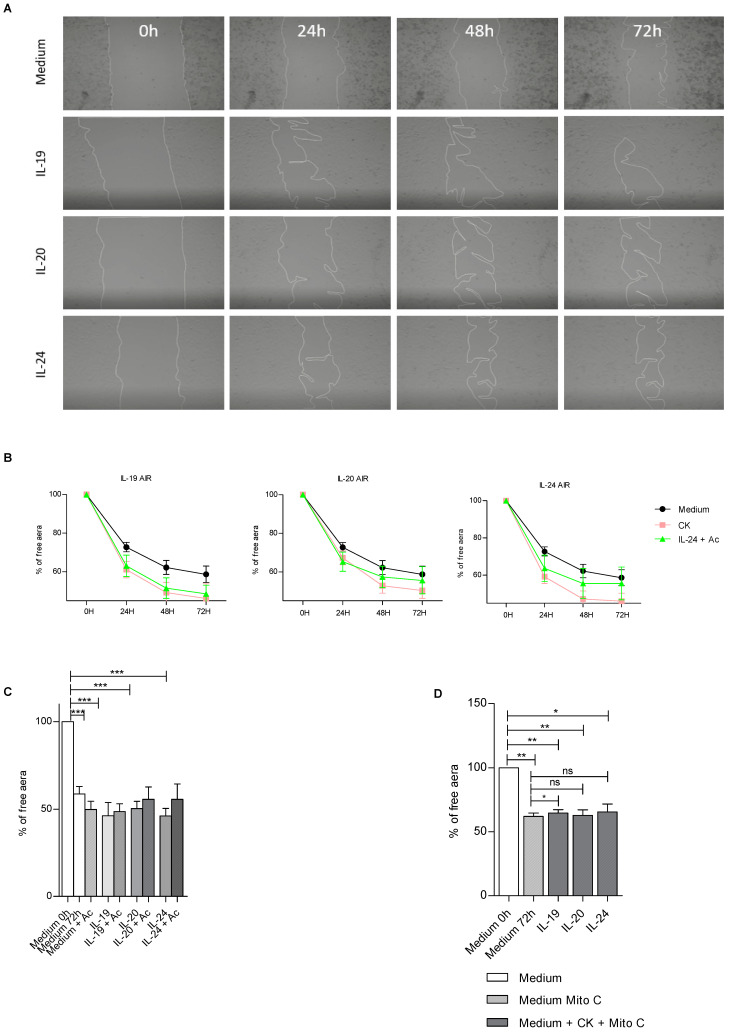
Modulation by IL-20 cytokines of epithelial repair in BEAS-2B cells in control conditions. (**A**) Representative pictures of epithelial repair of BEAS-2B cells, in the presence of IL-19, IL-20 or IL-24 cytokines, captured by phase-contrast microscopy (10× magnification). (**B**) Graphical representation of the cellular repair of BEAS-2B cells when treated or not with anti-IL-20Rb blocking antibody. “CK” means that BEAS-2B cells were independently treated with either exogenous cytokine IL19, IL-20 or IL-24, as indicated in the title of each plot in this part of the figure, and Ac means antibody for IL-20Rb. (**C**) Graphical representation at 72 h post-scratch of the cells in the presence of anti-IL-20RB antibody. (**D**) Role of IL-20 cytokine family on wound healing in control conditions and impact of the anti-proliferative agent, mitomycin C. Results are expressed as mean ± SEM and statistical analyses were performed by Mann–Whitney U analysis and *t*-test in comparison with cells not exposed to CSE. ns = non significant, * *p* < 0.05, ** *p* < 0.01 and *** *p* < 0.001. Independent experiments have been performed between 3–4 times.

**Figure 6 cells-12-02464-f006:**
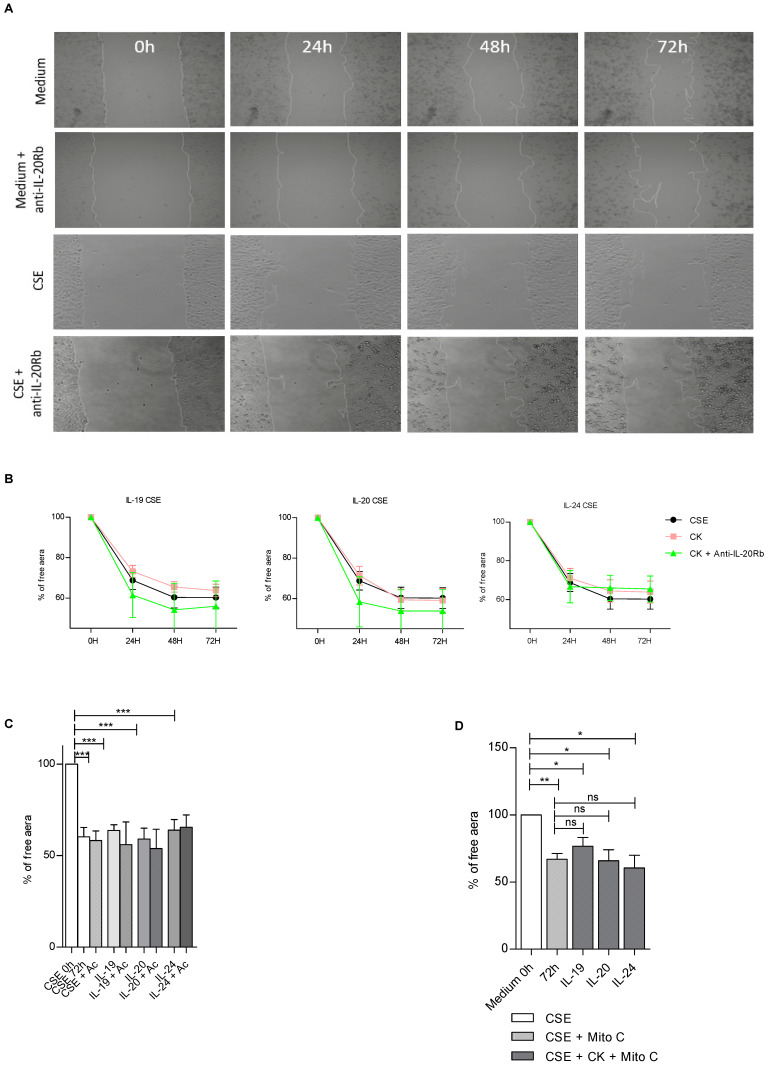
Modulation by IL-20 cytokines of epithelial repair in BEAS-2B cells in CSE conditions. (**A**) Representative pictures of epithelial repair of BEAS-2B cells treated with CSE, in the presence of IL-19, IL-20 or IL-24 cytokines, captured by phase-contrast microscopy (10x magnification). (**B**) Graphical representation of the cellular repair of BEAS-2B cells when treated with anti-IL-20Rb blocking antibody in CSE condition. “CK” means that BEAS-2B cells were independently treated with either exogenous cytokine IL19, IL-20 or IL-24, as indicated in the title of each plot in this part of the figure, and Ac means antibody for IL-20Rb. (**C**) Graphical representation at 72h post-scratch of the cells in the presence of anti-IL-20RB antibody and CSE treatments. (**D**) Role of IL-20 cytokine family on wound healing in CSE condition and impact of mitomycin C. Results are expressed as mean ± SEM and statistical analyses were performed by Mann–Whitney U analysis and *t*-test in comparison with cells not exposed to CSE. ns = non significant, * *p* < 0.05, ** *p* < 0.01 and *** *p* < 0.001. Independent experiments have been performed between 3–4 times.

**Table 1 cells-12-02464-t001:** Characteristics of patients. Data are expressed as mean ± SD or number (%). FEV1: Forced Expiratory Volume in one second. FVC: Forced Vital Capacity.

	Non COPD	COPD
*n*	26	20
Female/Male gender (%)	35/65	20/80
Age (years)	76 ± 9 [60–95]	72 ± 10 [54–93]
Smoking history		
*Never smokers (%)*	23	0
*Ex smokers (%)*	38.5	50
*Current Smokers (%)*	38.5	50
*Pack-years*	28 ± 24 [0–80]	50 ± 19 [20–100]
Spirometric GOLD Stage I/II	NA	10/10
Spirometry		
*FEV_1,_ % of predicted value*	97 ± 21 [71–165]	80 ± 15 [57–109]
*FVC, % of predicted value*	95 ± 17 [70–136]	95 ± 16 [62–124]
*FEV_1_/FVC*	0.80 ± 0.06 [0.72–0.92]	0.63 ± 0.06 [0.50–0.69]

**Table 2 cells-12-02464-t002:** Sequences of mouse primers used for RT-qPCR. Forward and reverse sequences of mouse primer used during RT-qPCR experiments are represented in this table.

Target	Primer	Sequence
*gapdh*	Forward	5′-TGTTTCCTCGTCCCGTAGACAA-3′
	Reverse	5′-GGCAACAATCTCCACTTTGCC-3′
*il-19*	Forward	5′-TGTGTGCTGCATGACCAACAA-3′
	Reverse	5′-GGCAATGCTGCTGATTCTCCT-3′
*il-20*	Forward	5′-TCTTGCCTTTGGACTGTTCTCC-3′
	Reverse	5′-GTTTGCAGTAATCACACAGCTTC-3′
*il-24*	Forward	5′-CCACTCTGGCCAACAACTTCAT-3′
	Reverse	5′-TCTGCGGAACAGCAAAAACC-3′
*il-20rb*	Forward	5′-CAGGTGCTTCCAGTCCGTCT-3′
	Reverse	5′-CTCTCCTGGAATCCCCAAAGT-3′

**Table 3 cells-12-02464-t003:** Sequences of human primers used for RT-qPCR. Forward and reverse sequences of human primer used during RT-qPCR experiments are represented in this table.

Target	Primer	Sequence
*actin*	Forward	5′-AGAGCTACGAGCTGCCTGAC-3′
	Reverse	5′-AGCACTGTGTTGGCGTACAG-3′
*il-19*	Forward	5′-ATCCAAGCTAAGGACACCTTCC-3′
	Reverse	5′-ATCCAAGCTAAGGACACCTTCC-3′ 5′-GTCACGCAGCACACATCTAAG-3′
*il-20*	Forward	5′-TTTTCTGAGATACGGGGCAGT-3′
	Reverse	5′-GTCTTAGCAAATGGCGCAGGA-3′
*il-24*	Forward	5′-CAACTGCAACCCAGTCAAGAAA-3′
	Reverse	5′-TGCTCTCCGGAATAGCAGAAA-3′
*il-20rb*	Forward	5′-TTCCACCTGGTTATTGAGCTGG-3′
	Reverse	5′-TGGAATACCCCCACTCCTCAC-3′

**Table 4 cells-12-02464-t004:** Histological scores. Mean ± SEM of the staining score with anti-IL-19, IL-20, IL-24 and IL-20Rb antibodies obtained in lung sections from patients. A semi-quantitative score (0 to 3) was established for large airways, distal airways, vessels, alveolar walls and inflammatory cells. * *p* < 0.05, ** *p* < 0.01, *** *p* < 0.001 versus non-smoker patients.

IL-19	Large Airway	Distal Airway	Vessel	Alveolar Wall	Inflammatory Cells
Non-Smoker	1.71 ± 0.18	1.71 ± 0.18	0.5 ± 0.15	0.93 ± 0.2	0.57 ± 0.3
Ex-Smoker	1.89 ± 0.2	1.89 ± 0.2	0.83 ± 0.17	1.01 ± 0.14	1.17 ± 0.24
Smoker	1.5 ± 0.26	1.6 ± 0.3	0.65 ± 0.13	0.85 ± 0.22	1.45 ± 0.27 *
COPD I	2 ± 0.22	2.1 ± 0.23	1.25 ± 0.23 **	1.6 ± 0.22 *	1.9 ± 0.23 ***
COPD II	2.33 ± 0.17	2.44 ± 0.18	1.44 ± 0.18 ***	1.67 ± 0.17 *	2.33 ± 0.24 ***
**IL-20**	**Large Airway**	**Distal Airway**	**Vessel**	**Alveolar Wall**	**Inflammatory Cells**
Non-Smoker	1 ± 0.31	0.79 ± 0.26	1 ± 0.05	0.36 ± 0.18	0.57 ± 0.13
Ex-Smoker	1.22 ± 0.21	1.33 ± 0.22	0.61 ± 0.11	0.83 ± 0.19	0.94 ± 0.06
Smoker	1.25 ± 0.21	1.25 ± 0.21	0.9 ± 0.16	0.9 ± 0.21	1.05 ± 0.23
COPD I	1.5 ± 0.17 **	1.6 ± 0.22 **	1.2 ± 0.13	1.3 ± 0.15 **	1.5 ± 0.22 **
COPD II	2 ± 0.21 ***	2.1 ± 0.23 ***	1.4 ± 0.16 *	1.6 ± 0.16 ***	1.8 ± 0.2 ***
**IL-24**	**Large Airway**	**Distal Airway**	**Vessel**	**Alveolar Wall**	**Inflammatory Cells**
Non-Smoker	0.79 ± 0.26	0.64 ± 0.26	0.57 ± 0.13	0.21 ± 0.1	0.07 ± 0.07
Ex-Smoker	1.22 ± 0.26	1.28 ± 0.24	0.72 ± 0.19	0.72 ± 0.21	0.89 ± 0.18 *
Smoker	1.8 ± 0.13 **	1.8 ± 0.13	1.2 ± 0.19 *	1.1 ± 0.16 *	1.7 ± 0.15 ***
COPD I	2 ± 0.29 ***	2 ± 0.29 **	1.5 ± 0.2 **	1.61 ± 0.31 ***	1.89 ± 0.31 ***
COPD II	2.67 ± 0.17 ***	2.67 ± 0.17 ***	2.33 ± 0.18 ***	2.44 ± 0.24 ***	2.45 ± 0.24 ***
**IL-20Rb**	**Large Airway**	**Distal Airway**	**Vessel**	**Alveolar Wall**	**Inflammatory Cells**
Non-Smoker	0.86 ± 0.21	0.86 ± 0.21	0.36 ± 0.14	0.21 ± 0.15	0.29 ± 0.18
Ex-Smoker	0.83 ± 0.12	0.83 ± 0.12	0.44 ± 0.06	0.67 ± 0.12	1 ± 0.14 *
Smoker	1 ± 0.16	1 ± 0.13	0.6 ± 0.1	0.35 ± 0.11	1.05 ± 0.17 **
COPD I	1.78 ± 0.15 **	1.78 ± 0.15 **	0.94 ± 0.18 **	1.39 ± 0.2 ***	1.83 ± 0.17 ***
COPD II	2 ± 0.24 ***	2 ± 0.17 ***	1.33 ± 0.17 ***	1.78 ± 0.22 ***	2.11 ± 0.2 ***

## Data Availability

All data are available upon request.

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
