# Peer review of "IL-20 Cytokines Are Involved in the Repair of Airway Epithelial Barrier: Implication in Exposure to Cigarette Smoke and in COPD Pathology"

_cells, 2023, doi:10.3390/cells12202464_

Round 1
Reviewer 1 Report
Barada and colleagues used in vitro and ex vivo systems to investigate the role of interleukin-20 (IL-20) cytokines and its associated receptors in the repair of airway epithelial barrier and in early COPD pathology. The study was well-conducted. The authors were commended for highlighting the potential implications of IL-20 cytokines in airway repair and an antibody against IL-20 receptor (IL20Rb) as a future therapeutic strategy for COPD. However, several major aspects remain to be addressed in order for the article to be considered for publication in the Cells.
Major comments:
1. The authors collected human lung tissues to establish a cohort, which is the major strength of the study. Did the authors observe any changes in the expression of IL-20 cytokines and their receptors in the other compartments (alveolar septa or pulmonary vasculature)?
2. In line with the comment above, studies have shown that female sex hormones may account for greater COPD susceptibility, especially the remodelling in small airways (DOI: 10.1164/rccm.201512-2379ED). Do the authors observe any difference in the context of IL-20 cytokines family members (and their receptors) between males and females?
3. The authors performed wound healing assays using airway epithelial cells to investigate the impact of IL-20 cytokines on epithelial cell repair. Based on the results, cigarette smoke (CS) altered the dynamic of IL-20 cytokines-mediated epithelial barrier repair observed in medium control groups. A particularly intriguing observation was the expedited barrier repair when an anti-IL-20Rb antibody was introduced in CS-exposed epithelial cells. Could IL-20 cytokines bind to other receptors, initiating the repair process, when IL-20Rb is blocked in cells? Could the authors please perform an experiment in the cells to determine whether the expression of other IL-20 cytokines binding receptors is altered upon IL-20Rb blockade?
4. In line with the question above, the authors nicely showed that IL-20 cytokines increased epithelial repair and pointed out the connection between IL-20 and MMPs. Given that one of the main features in COPD is small airway fibrosis, could the authors please investigate the expression of EMT markers (Vimentin, E-cadherin, N-cadherin…) in both medium-exposed and CS extra-exposed epithelial cells?
5. In figure 2B, IL-20Rb expression was significantly increased in CS-exposed mouse lungs, but it was nearly the same level in tracheal epithelial cells (mTEC) from Air- or CS-exposed mice in Figure 4B. Are these cells comparable with human BEAS-2B cells?
If so, could the authors please address the potential discrepancy in the results obtained from human BEAS-2B and mTEC? If not, could the authors please explain the choice of the cells in the study and its potential impact on the expression of IL-20Rb?
Minor comments:
1. Could the authors provide the number of ethical approvals for this study (I have only seen the approval date)?
2. Please italicize the species and gene symbol in the article. For instance, in line 216, Mus musculus or in line 303, encoding for IL-19.
3. In figure legend 2A, the author wrote “Expression of IL-19, IL-20, and IL-24 was evaluated…”. However, in the figure there were 4 investigated targets. Could the authors double-check again if there was a mistake?
4. Line 255, smocking status, should be “smoking” and line 428, non”-“COPD (to be consistent as line 430).
5. The manuscript sometimes uses “il-19” and other times “IL-19”. Please clarify if there is a specific reason for the variations or maintain consistency throughout.
6. Could the authors justify in what condition non-parametric Mann-Whitney U analysis was used and in what condition parametric Student’s t-test was used?
7. Please provide the full forms of abbreviations used in the figures in the figure legends. In particular, Figures 5 and 6 (e.g. Ac, CK, Mito C…). Is the Figure 5B, IL-24 + Ac, the same condition as CK + Anti-IL20Rb? If so, please choose one to stay consistent.
8. Although magnification was included in the figure legends, the authors must include scale bars in all images obtained from the microscope (Figures 1, 2, 5 and 6). Please also indicate the length of the scale bars.
9. In line 257, the authors wrote, “Regarding IL-20Rb, this subunit is…cell type including BEC within the airway epithelium.” Could the authors show an enlarged image focusing on what was described in the article? Alternatively, the authors could use a symbol (arrows, arrowheads…) to point out.
N/A
Author Response
Reviewer 1
Barada and colleagues used in vitro and ex vivo systems to investigate the role of interleukin-20 (IL-20) cytokines and its associated receptors in the repair of airway epithelial barrier and in early COPD pathology. The study was well-conducted. The authors were commended for highlighting the potential implications of IL-20 cytokines in airway repair and an antibody against IL-20 receptor (IL20Rb) as a future therapeutic strategy for COPD. However, several major aspects remain to be addressed in order for the article to be considered for publication in the Cells.
Major comments:
- The authors collected human lung tissues to establish a cohort, which is the major strength of the study. Did the authors observe any changes in the expression of IL-20 cytokines and their receptors in the other compartments (alveolar septa or pulmonary vasculature)?
As observed on lung sections, endothelial cells are also positive for IL-20 cytokine staining, as well as for their receptors. It seems therefore that both epithelial and endothelial cells are sources of pulmonary IL-20 cytokines. In addition, we also observed that immune cells that can infiltrate the lung compartment, and especially macrophages, express the subunit of IL-20 cytokine receptors. Such comments have been added to the original text.
- In line with the comment above, studies have shown that female sex hormones may account for greater COPD susceptibility, especially the remodelling in small airways (DOI: 10.1164/rccm.201512-2379ED). Do the authors observe any difference in the context of IL-20 cytokines family members (and their receptors) between males and females?
We agree with this comment. Sex is known to influence the disease features. However, we did not perform our analysis according to this criteria. Further studies are needed to clarify if sex could play a role on the IL-20 cytokine axis.
- The authors performed wound healing assays using airway epithelial cells to investigate the impact of IL-20 cytokines on epithelial cell repair. Based on the results, cigarette smoke (CS) altered the dynamic of IL-20 cytokines-mediated epithelial barrier repair observed in medium control groups. A particularly intriguing observation was the expedited barrier repair when an anti-IL-20Rb antibody was introduced in CS-exposed epithelial cells. Could IL-20 cytokines bind to other receptors, initiating the repair process, when IL-20Rb is blocked in cells? Could the authors please perform an experiment in the cells to determine whether the expression of other IL-20 cytokines binding receptors is altered upon IL-20Rb blockade?
Such experiments have been performed but have not been included in the manuscript. We observed that IL-20Rb blockade reduced the effects of IL-20 cytokines on wound healing process. Since the take-home message of the manuscript is that CS exposure triggers IL-20 cytokine upregulation and these mediators are responsible for the defect in wound healing, we decided to not add the control experiment using the anti-IL20Rb Ab and exogenous IL-20 CK.
- In line with the question above, the authors nicely showed that IL-20 cytokines increased epithelial repair and pointed out the connection between IL-20 and MMPs. Given that one of the main features in COPD is small airway fibrosis, could the authors please investigate the expression of EMT markers (Vimentin, E-cadherin, N-cadherin…) in both medium-exposed and CS extra-exposed epithelial cells?
We totally agree with this comment. We performed some experiments related to the expression of molecules involved in tight junctions. As shown on the figure below (n=6), it seems that exogenous IL-24 is able to upregulate the expression of Occludin and E-Cadherin, but only in control condition (Air). After CS exposure (COPD condition), no impact of recombinant cytokine was observed on the expression of tight-junction molecules.
In addition, such parameters have been previously described by our team. In vivo IL-20 cytokines have an impact on tight junction and play a role in epithelial lesions (IL-20 Cytokines Are Involved in Epithelial Lesions Associated with Virus-Induced COPD Exacerbation in Mice, Biomedicines 2021). In the litterature, it has also been reported that IL-22, a cytokine belonging to the same family as the IL-20 cytokines, increases permeability of intestinal epithelial tight junctions by enhacing claudin-2 expression (Wang et al., J Immunol 2017 ; Tsai et al., Cell Host Microbe 2017).
- In figure 2B, IL-20Rb expression was significantly increased in CS-exposed mouse lungs, but it was nearly the same level in tracheal epithelial cells (mTEC) from Air- or CS-exposed mice in Figure 4B. Are these cells comparable with human BEAS-2B cells? If so, could the authors please address the potential discrepancy in the results obtained from human BEAS-2B and mTEC? If not, could the authors please explain the choice of the cells in the study and its potential impact on the expression of IL-20Rb?
Highly regulated programs for airway epithelial cell proliferation and differentiation during development and repair are often disrupted in disease. These processes have been studied in mouse models; however, it is difficult to isolate and identify epithelial cell-specific responses in vivo. To investigate these processes in vitro, primary cultures of mouse tracheal epithelial cells (mTEC) are commonly used. Genetically altered or injured mouse tracheal epithelial cells (mTEC) also reflected in vivo patterns of airway epithelial cell gene expression (You et al., Lung Cell Mol Physiol 2002 ; Horani et al., Springer Protocols 2013).
Mouse and human cells are not comparable. mTEC are mouse tracheal epithelial cells whereas BEAS-2B cells are human bronchial epithelial cells. mTEC and BEAS-2B do not represent the same part of the airways. These ones represent in vitro models, which can always be improved. But these ones are used as reference to study lung pathologies. In our case, there is an imprinting of CS effects on mTEC. In contrast, exogenous impact of CS was observed on BEAS-2B cells. In both cases, CS triggered IL-20 production.
Another reason with we used mTEC to investigate in vitro mechanisms is that we tried to follow the 3R rules at their maximum. To limit the number of mice used for our in vivo experiments, we decided to dedicate the lung tissue to histology, ELISA and PCR and to use trachea to obtain our ex vivo model.
Minor comments:
- Could the authors provide the number of ethical approvals for this study (I have only seen the approval date)?
All the numbers have been added in the manuscript: human work has been mentionned.
Paraffin-embedded sections of the human specimens used in this study were obtained from the pathology department of the Reims University Hospital as part of the inclusion of these specimens in the Reims University Hospital Biological Resource Center collection.
The Biological Resource Center collection is a collection declared and authorized by the French Ministry of Health (N°-2008-374).
Under French law, the surgeon is responsible for informing patients and obtaining their consent (see enclosed forms), and the Biological Resource Center is responsible for collecting consent forms from surgeons, for collecting associated data (such as age, sex, smoking status, etc.) and for anonymization of the consent forms.
Without consent, samples are not included in the collection.
- Please italicize the species and gene symbol in the article. For instance, in line 216, Mus musculusor in line 303, encoding for IL-19.
As suggested, changes have been made throughout the manuscript.
- In figure legend 2A, the author wrote “Expression of IL-19, IL-20, and IL-24 was evaluated…”. However, in the figure there were 4 investigated targets. Could the authors double-check again if there was a mistake?
The reviewer is right, a mistake was made. More targets have been investigated.
- Line 255, smocking status, should be “smoking” and line 428, non”-“COPD (to be consistent as line 430).
This has been modified according to the comment. All these groups of patients have been discriminated for our analysis: non-smokers, smokers, ex-smokers, GOLD I and II COPD patients.
- The manuscript sometimes uses “il-19” and other times “IL-19”. Please clarify if there is a specific reason for the variations or maintain consistency throughout.
This has been modified. Yes, there's a reason for the typeface. For example, when we write IL-19, we're talking about the protein, and when we write il-19, we're talking about the gene. This is the commonly used nomenclature.
- Could the authors justify in what condition non-parametric Mann-Whitney U analysis was used and in what condition parametric Student’s t-test was used?
Firstly, a Mann-Whitney U test is used to validate the statistics of the whole experiment, as working with live animals, cells…, our samples don't necessarily follow the binomial distribution, which is why we use this non-parametric statistical test. Subsequently, Student's t-test is only used for independent 2-to-2 condition comparison to determine their potential difference.
- Please provide the full forms of abbreviations used in the figures in the figure legends. In particular, Figures 5 and 6 (e.g. Ac, CK, Mito C…). Is the Figure 5B, IL-24 + Ac, the same condition as CK + Anti-IL20Rb? If so, please choose one to stay consistent.
As suggested, modifications have been done to stay consistent.
- Although magnification was included in the figure legends, the authors must include scale bars in all images obtained from the microscope (Figures 1, 2, 5 and 6). Please also indicate the length of the scale bars.
As suggested, scale bars have been added on figures.
- In line 257, the authors wrote, “Regarding IL-20Rb, this subunit is…cell type including BEC within the airway epithelium.” Could the authors show an enlarged image focusing on what was described in the article? Alternatively, the authors could use a symbol (arrows, arrowheads…) to point out.
Special signs have been integrated onto the pictures to point out specific cell types.
Reviewer 2 Report
This study by Barada and co-workers explore the role of IL-20 and related mediators on epithelial repair in COPD by a combination of experiments using COPD lung tissues, in vivo mice experiments and in vitro epithelial cell experiments. They conclude that IL-20 and it related cytokines impact epithelial repair in COPD and suggest that modulation of the IL-20 receptor on airway epithelial cells could be a potential therapeutic pathway to target in COPD to alter epithelial repair.
Comments:
1) This reviewer has an issue with the overall design of the study to address this hypothesis
a. The immunohistochemical analysis of the human lung tissues is subjective. There is no mention of how the samples to analyse were selected. Were there any form of randomization in selecting what images were analyzed or what specific structures were analyzed (larger airways, smaller airways and what size, lung parenchyma or alveolar structure.
b. Why not use morphometric methods and image analysis programs to quantify staining of airways epithelium. That will give at least a more quantitative analysis of the expression of IL-20 and related mediators on airways epithelial tissues
c. Authors mention they use lung injury score but give no reference or results from applying the score
d. Seeing that the authors have access to human lung tissues from COPD and non-COPD patients, why not using primary epithelial cells from these tissues to do the in vitro epithelial cell injury experiments rather than BAES cells. This will give much more valuable information about the role of IL-20 in COPD epithelial repair.
e. In mice experiments the authors use tracheal epithelial cells for analysis. COPD is predominantly a disease of smaller airways and epithelial cells in the tracheal and smaller airways are distinctly different. The results from these trachea cell most likely does not represent pathological processes in smaller airways.
2) Results:
a. Figure 1A: The author mentioned that they use representative pictures. How was that selected to reduce bias. The conclusions make from these pictures need to be backed up by quantitative or at least semi-quantitative analysis of the immunohistochemical staining
b. Pictures of smaller and larger airways are shown but more convincing will be to use pictures of the same small airway stained for the different mediators/receptor throughout
c. Figure 2a the expression of the receptors between controls and CS exposure looks similar on immunohistochemical pictures provided. Again why not quantify the staining with image analysis techniques
d. Figure 3 showed increase IL-24 in BAES cells which is interesting seeing that these cytokines are made mostly by hematopoietic cells such as monocytes, lymphocytes etc. This need to be explained and discussed
e. Figure 5&6 one of the groups is “CK” but the authors does not define what “CK stand for?
3) Clearly exposure of mice and standard bronchial epithelial cells to CS is different from COPD in humans. As most of the quantitative results of this paper relate to CS experiments the authors need to change the title of the MS to reflect that.
Reviewer 3 Report
This paper demonstrated that IL-20 cytokines may promote airway epithelial barrier repair in COPD. The introduction is well written and flows well. There are major concerns with figures in the results section as images are blurry. It is therefore difficult to tell whether descriptions and interpretations are accurate. Some issues are listed below:
Methods:
· Table 1- Non COPD Ex smokers (%) and Current smokers (%)- replace comma with decimal point
· Lung immunohistochemistry- More information require. What was the scoring criteria? Blinded? How were images take (microscope, magnification, etc)?
· Cytokine measurement by ELISA- States that cytokine levels measured in BEAS-2B cell lysates, were levels also measured in human lung samples?
Results:
· Please present figures in a clearer format. Many of the images (particularly graphs and tables) are difficult to read.
· Difficult to tell whether images are representative without quantification
· Are fold changes expressed as the average value of all control groups? Can individual values obtained for control groups be expressed as the average value of all control groups? How was analysis performed?
Round 2
Reviewer 1 Report
The current form of the manuscript is satisfactory.
Author Response
We thank Reviewer 1 who help us to greatly improve our manuscript.
Reviewer 2 Report
All questions and comments satisfactory addressed
Satisfactory
Author Response
We thank Reviewer 2 for his comments that improved the quality of the manuscript
Reviewer 3 Report
This paper demonstrated that IL-20 cytokines may promote airway epithelial barrier repair in COPD. While authors have addressed some of the comments made by reviewers, there is still a major concern with data presentation. Due to the low quality, it is difficult to confirm whether descriptions and interpretations are accurate. Please consult with the journal if this is not what you are seeing on your end.
Quality of English is adequate.
Author Response
We thank Reviewer 3 comments. A new version of Figure 1 with better definition has been added to the manuscript.